# Prognostic model of in-hospital ischemic stroke mortality based on an electronic health record cohort in Indonesia

Nizar Yamanie[1,2], Yuli Felistia[2], Nugroho Harry Susanto[3], Aly Lamuri[2], Amal Chalik Sjaaf[4], Muhammad Miftahussurur[5], Anwar Santoso[6]*

1 Doctoral Program of Medical Science, Faculty of Medicine, Airlangga University, Surabaya, Indonesia, 2 National Brain Centre Hospital, Jakarta, Indonesia, 3 Indonesia Research Partnership on Infectious Diseases (INA-RESPOND), Jakarta, Indonesia, 4 Department of Public Health, University of Indonesia, Jakarta, Indonesia, 5 Division of Gastroentero-Hepatology, Department of Internal Medicine, Faculty of Medicine-Dr. Soetomo Teaching Hospital, Airlangga University, Surabaya, Indonesia, 6 Department of Cardiology–Vascular Medicine, National Cardiovascular Centre–Harapan Kita Hospital, Universitas Indonesia, Jakarta, Indonesia

* anwarsantoso@inaheart.org

**Data Availability Statement:** All relevant data are within the manuscript and its Supporting Information files.

## Abstract

Stroke patients rarely have satisfactory survival, which worsens further if comorbidities develop in such patients. Limited data availability from Southeast Asian countries, especially Indonesia, has impeded the disentanglement of post-stroke mortality determinants. This study aimed to investigate predictors of in-hospital mortality in patients with ischemic stroke (IS). This retrospective observational study used IS medical records from the National Brain Centre Hospital, Jakarta, Indonesia. A theoretically driven Cox's regression and Fine-Gray models were established by controlling for age and sex to calculate the hazard ratio of each plausible risk factor for predicting in-hospital stroke mortality and addressing competing risks if they existed. This study finally included 3,278 patients with IS, 917 (28%) of whom had cardiovascular disease and 376 (11.5%) suffered renal disease. Bivariate exploratory analysis revealed lower blood levels of triglycerides, low density lipoprotein, and total cholesterol associated with in-hospital-stroke mortality. The average age of patients with post-stroke mortality was 64.06 ± 11.32 years, with a mean body mass index (BMI) of 23.77 kg/m² and a median Glasgow Coma Scale (GCS) score of 12 and an IQR of 5. Cardiovascular disease was significantly associated with IS mortality risk. NIHSS score at admission (hazard ratio [HR] = 1.04; 95% confidence interval [CI]: 1.00–1.07), male sex (HR = 1.51[1.01–2.26] and uric acid level (HR = 1.02 [1.00–1.03]) predicted survivability. Comorbidities, such as cardiovascular disease (HR = 2.16 [1.37–3.40], pneumonia (HR = 2.43 [1.42–4.15] and sepsis (HR = 2.07 [1.09–3.94], had higher hazards for post-stroke mortality. Contrarily, the factors contributing to a lower hazard of mortality were BMI (HR = 0.94 [0.89–0.99]) and GCS (HR_eye = 0.66 [0.48–0.89]. In summary, our study reported that male sex, NIHSS, uric acid level, cardiovascular diseases, pneumonia, sepsis. BMI, and GCS on admission were strong determinants of in-hospital mortality in patients with IS.

**Funding:** The author(s) received no specific funding for this work.

**Competing interests:** The authors have declared that no competing interests exist.

## Introduction

Stroke or cerebrovascular disease arises from an ischemic or hemorrhagic etiology. Globally, stroke contributed to 3.54% of all-cause mortality in 1990 and 4.62% in 2013 [1]. Stroke incidence, prevalence, mortality, and disability-adjusted life-year (DALY) rates increase from 1990 to 2013. However, the overall stroke burden, in terms of the absolute number of people affected by or who remained disabled from stroke, has escalated worldwide in both men and women of all ages [2].

Based on the global burden of disease, disability caused by stroke is steadily increasing, as denoted by the remaining high absolute number of DALY [3]. Although the Southeast Asian (SEA) countries do not have a complete report on prevalence and incidence rates, the mortality rate is 56–193.3/100,000 person-years. Indonesia, Myanmar, and Lao PDR are ranked as the countries with the highest IS-related mortality rates in SEA (193.3, 165.4, and 141.3 /100,000 person-years respectively) [4]. Within one month, the case fatality rate (CFR) for IS is 13.5% worldwide and 10.8% in Asia [5].

Identifying the mortality determinants is imperative for appropriately profiling the risk in patients with IS. Several vascular risk factors contribute to the occurrence of stroke and recurrent stroke, including hypertension, diabetes, hypercholesterolemia, and smoking [6]. Patients with diabetes who present with stroke have a higher mortality risk, with a hazard ratio of 2.15 [7]. Hypertension, chronic heart failure, ischemic cardiovascular disease, and atrial fibrillation also contribute to post-stroke mortality [8]. Prior to mortality, a lower health-related quality of life (HRQoL) is also observed among older patients of non-Asian ethnicity with higher National Institutes of Health Stroke Scale (NIHSS) scores. To improve HRQoL, it is paramount to implement improved stroke care, which in turn decreases mortality and CFR [9], especially during hospital care.

However, we are yet to completely unravel the determinants of post-stroke mortality in SEA, particularly in hospital settings. A study in Taiwan reported rates of in-hospital mortality in ischemic stroke of 6–8% and in Asia of around 8.8% [10]. The first publication regarding in-hospital stroke mortality from Kazakhstan with a population from central Asia shows that hypertension, chronic heart failure, ischemic heart disease, and atrial fibrillation are the main risk factors [8]. Furthermore, a study in Southwestern Saudi Arabia shows that the in-hospital mortality rate is influenced by pre-stroke smoking, pre-stroke hypertension, poor mobility, post-stroke disturbed consciousness, and pulmonary embolism [11]. A previous publication highlighted the varying degrees of outcome affected by stroke. In Indonesia, there is limited data available regarding poststroke mortality in hospital care.

The National Brain Centre Hospital (NBCH) is a tertiary care hospital that provides medical care for neurological diseases, including stroke. It was first established by the Ministry of Health of the Republic of Indonesia to anticipate a steadily increasing prevalence of stroke. Since 2013, the NBCH has been the national reference for neurological cases, contributing to the annual management of at least 3,000 IS patients.

Providing holistic post-stroke care requires a thorough investigation of the factors that influence stroke clinical outcomes, including mortality. This study aims to determine the hazard factors and survivability in ischemic stroke among a cohort of Indonesian patients.

## Methods

### Study design and eligibility criteria

This retrospective, observational study was conducted at the NBCH in Jakarta, Indonesia. All patients underwent a standardized neurological assessment at hospital admission. The study

population included hospitalized IS patients according to the Trial of Org 10172 in Acute Stroke Treatment (TOAST) definition along with the ICD-10 code I63 for acute ischemic stroke [12]. Patients with intracranial hemorrhage (ICH), subarachnoid hemorrhage, or transient ischemic attack (TIA) were excluded. Clinical assessment included information on demographic characteristics, personal and family histories, and vascular risk factors. Neurological examination, biochemical blood tests, and computed tomography or magnetic resonance imaging of the brain were performed in all patients as per our national standard at admission. Stroke severity at baseline was assessed using the NIHSS, and consciousness level was assessed using the Glasgow Coma Scale (GCS) [13]. Verbal incoherence was defined as a verbal component of the GCS score of $\leq 4$.

Cardiovascular diseases were determined using the International Classification of Diseases, 10th revision (ICD-10), namely, diseases of the circulatory system I00-I99. Ischemic heart disease was defined as hospital admission with ICD-10 codes I20-I25, hypertensive heart disease with ICD-10 codes I10-I16, and diseases of the arteries, arterioles, and capillaries with ICD-10 codes I70-I79.

Renal disease was defined as declining glomerular function, as indicated by a lower estimated glomerular filtration rate (eGFR) of $\leq 90$ mL/min/1.73 m$^2$. As a terminal phase of renal disease, renal failure was recorded as ICD-10 codes N17-N19. The cause of renal failure was defined by ICD-10 codes N00-N08 for glomerular diseases, N10-N16 for renal tubulo-interstitial diseases, N20-N23 for urolithiasis, and N25-N29 for other disorders of the kidney and ureter. All methods were performed in accordance with the STROBE and RECORD statements [14, 15].

We defined uncontrolled diabetes and uncontrolled hypertension per ICD10 terms. Uncontrolled diabetes was defined as a patient with diabetes mellitus with uncontrolled HbA1 level > 8%. [16]. Uncontrolled hypertension definition was determined as a patient with untreated hypertension or not responding to standard current treatment [17].

## Data collection and sources

Data for a period of one year was collected from 1 January to 31 December 2020. The manually collected registry was derived from the electronic medical records in the hospital. The collected data comprised sociodemographic and clinical variables, where each row represented a patient, and each column represented the variable of interest. The patient's discharge status was recorded in the medical records. This study included all the recorded IS cases, and it was approved by the institutional review board or independent ethics committee of the NBCH (No: LB.02.01/KEP/089/2021). As the risk to the participants of the study was not greater than the minimum risk in medical care, the requirement for individual written informed consent was waived. The dataset contained no variables that could be used to identify patients.

## Clinical outcomes

The clinical outcomes were all-cause and stroke-related mortality rates during hospitalization. All the patients were monitored during their hospitalization. For patients with in-hospital mortality, the cause of death was reviewed by a registered neurologist.

## Statistical analysis

All statistical analyses used R 4.1.3 (R Foundation for Statistical Computing, Vienna, Austria). An exploratory analysis was performed to describe the clinical variables of the study sample across each pertinent variable. The complete dataset was separated into numerical and categorical variables, and entries with missing data were removed during analysis. The numeric

dataset was examined for centrality and spread, grouped by the outcomes of being alive (no event) and death (event). Theoretically driven models explaining mortality in IS were explored in a multivariable analysis (Fig 1A). Plausible factors considered to affect clinical outcomes in stroke patients were as follows: history of uncontrolled diabetes or hypertension; [16, 17] the presence of cardiovascular disease or renal disease; [18] patient age and sex; [9, 19–21] blood pressure, measured as systole and diastole; GCS separated into individual measures of the eye (E), movement (M), and verbal (V); body mass index (BMI); and blood level of uric acid. To aid the exploratory procedure, significant variables reported in previous studies are included in stepwise regression models, using a combination of forward and backward selection (Fig 1A). All variables selected through the stepwise regression were then selected as variables of interest in a time-to-event analysis [22].

Besides variables selected through the stepwise procedure, other multivariable models were further examined during the exploratory phase. All classification models were compared using classification metrics, and only the best-performing model was interpreted. The final model included the following variables to adjust the estimation of the hazard ratio: age, sex, cardio-vascular disease, renal disease, BMI, uric acid, NIHSS and GCS components. Identified risk factors were used to estimate the hazard of in-hospital-stroke mortality using a Cox's regression and Fine-Gray model, where the outcomes were alive (0) and deceased (1) and the time was set on a daily increment. The adjusted hazard ratio (HR) was calculated by exponentiating the estimated values for each variable. The Fine-Gray model and cumulative incidence function (CIF) were used in addition to the Kaplan-Meier survival analysis to account for the competing risk of in-hospital mortality. All statistical analysis steps were reported as S1 File.

## Results

The complete dataset included 3,561 entries, with 167 (4.69%) recorded mortalities. Around 14% of the patients (499 out of 3,561) were referred from other previous hospitals. The top three causes of death were cardiac arrest, pneumonia, and sepsis. Table 1 describes the statistical measures of centrality and spread. Upon missing data removal, our analysis retained 3,278 entries (Fig 1B) (7.9% missing data removed), consisted of 114 (3.48%) mortalities; 917 patients (28%) had cardiovascular disease, and 376 (11.5%) had renal disease.

Interestingly, exploratory analysis suggested that patients with post-stroke mortality had low blood triglyceride levels (126 ± 67.55 vs. 153 ± 110.78 mg/dL). A similar trend was observed for low density lipoprotein (LDL) levels (112 ± 51.49 vs. 134 ± 42.81 mg/dL; for dead and alive patients, respectively) and total cholesterol (172 ± 54.85 vs. 193 ± 51.21 mg/dL) levels, but not for high density lipoprotein (HDL) levels (41 ± 13.74 vs. 40 ± 13.46 mg/dL) (Table 1). Blood glucose levels would also be another variable of interest, but they were not included due to the presence of missing values in the event group. Of all the recorded entries, the missing values were 25%, 36%, and 31% for fasting blood glucose, postprandial blood glucose, and HbA1C, respectively.

### Clinical profiles of patient groups

The characteristics of the patients in Table 1 showed that older subjects and patients with a lower BMI had worse outcomes. The patients' ages were 64.06 ± 11.32 years and 59.05 ± 10.93 years in the event and no event groups, respectively. The BMI was 23.77 ± 3.68 kg/m$^2$ in the event group and 25.11 ± 3.84 kg/m$^2$ in the non-event group. Irrespective of the group, patients included in all analysis were 59.23 ± 10.98 years old, with a BMI of 25.07 ± 3.84 kg/m$^2$. The median ± inter quartile range of GCS scores were 12 ± 5 and 15 ± 0 in the event and no event groups, respectively.

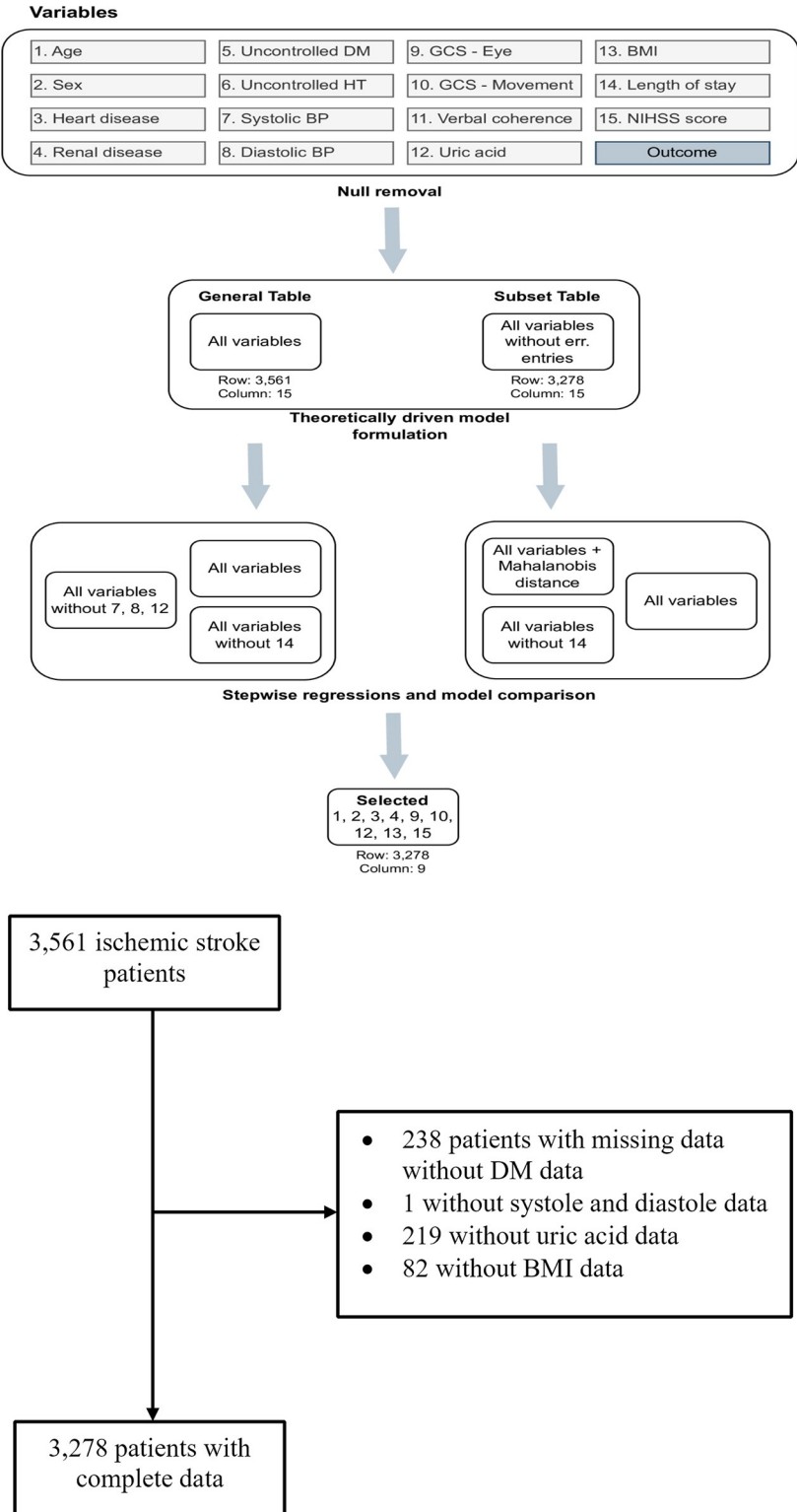

**Fig 1. A.** Variables selection procedure. **B.** Patients selection flow chart.

**Table 1. Baseline characteristics of ischemic stroke patients by event groups.**

| | Overall | Grouped by Event | | | | |
|---|---|---|---|---|---|---|
| Characteristic | N = 3,278 | Alive, N = 3,164 | Deceased, N = 114 | Difference | 95% CI | p-value |
| Age (years) | 59.23 [10.98] | 59.05 [10.93] | 64.06 [11.32] | -5.0 | -7.1, -2.9 | <0.001 |
| Cardiovascular disease (yes/no) | 917 (28%) | 843 (27%) | 74 (65%) | -38% | -48%, -29% | <0.001 |
| Renal disease (yes/no) | 376 (11%) | 334 (11%) | 42 (37%) | -26% | -36%, -17% | <0.001 |
| Uncontrolled diabetes (yes/no) | 1,084 (33%) | 1,038 (33%) | 46 (40%) | -7.5% | -17%, 2.1% | 0.11 |
| Uncontrolled hypertension (yes/no) | 2,256 (69%) | 2,179 (69%) | 77 (68%) | 1.3% | -7.9%, 11% | 0.8 |
| Systole (mmHg) | 157.49 [27.80] | 157.61 [27.78] | 154.14 [28.26] | 3.5 | -1.9, 8.8 | 0.2 |
| Diastole (mmHg) | 89.69 [14.80] | 89.81 [14.75] | 86.25 [16.00] | 3.6 | 0.56, 6.6 | 0.021 |
| GCS—Eye | | | | 0.00 | | <0.001 |
| 0 | 1 (<0.1%) | 1 (<0.1%) | 0 (0%) | | | |
| 1 | 13 (0.4%) | 6 (0.2%) | 7 (6.1%) | | | |
| 2 | 44 (1.3%) | 35 (1.1%) | 9 (7.9%) | | | |
| 3 | 181 (5.5%) | 150 (4.7%) | 31 (27%) | | | |
| 4 | 3,038 (93%) | 2,971 (94%) | 67 (59%) | | | |
| 5 | 1 (<0.1%) | 1 (<0.1%) | 0 (0%) | | | |
| GCS—Movement | | | | 0.00 | | <0.001 |
| 0 | 1 (<0.1%) | 1 (<0.1%) | 0 (0%) | | | |
| 1 | 6 (0.2%) | 4 (0.1%) | 2 (1.8%) | | | |
| 2 | 4 (0.1%) | 2 (<0.1%) | 2 (1.8%) | | | |
| 3 | 5 (0.2%) | 4 (0.1%) | 1 (0.9%) | | | |
| 4 | 33 (1.0%) | 22 (0.7%) | 11 (9.6%) | | | |
| 5 | 259 (7.9%) | 217 (6.9%) | 42 (37%) | | | |
| 6 | 2,970 (91%) | 2,914 (92%) | 56 (49%) | | | |
| Verbal coherence (%) | 2,819 (86%) | 2,766 (87%) | 53 (46%) | 41% | 31%, 51% | <0.001 |
| Uric acid (mg/dl) | 5.97 [4.43] | 5.95 [4.49] | 6.56 [2.33] | -0.61 | -1.1, -0.15 | 0.009 |
| BMI (kg/m$^2$) | 25.07 [3.84] | 25.11 [3.84] | 23.77 [3.68] | 1.3 | 0.64, 2.0 | <0.001 |
| LoS (days) | 5.09 [3.39] | 4.98 [3.15] | 8.13 [6.85] | -3.1 | -4.4, -1.9 | <0.001 |
| NIHSS (score) | 6.42 [4.75] | 6.22 [4.60] | 12.01 [5.58] | -5.8 | -6.8, -4.7 | <0.001 |
| RBS (mg/dl) | 166.81 [89.66] | 166.66 [89.80] | 170.98 [86.10] | -4.3 | -21, 12 | 0.6 |
| Pneumonia (n/%) | 220 (6.7%) | 162 (5.1%) | 58 (51%) | -48% | -58%, -38% | <0.001 |
| Sepsis (n/%) | 27 (0.8%) | 10 (0.3%) | 17 (15%) | -15% | -23%, -8.0% | <0.001 |
| Trigliceryde (mg/dl) | 152.68 [112.24] | 153.67 [113.32] | 124.92 [71.42] | 29 | 15, 43 | <0.001 |
| LDL-C (mg/dl) | 133.69 [43.37] | 134.34 [42.87] | 115.67 [52.76] | 19 | 8.5, 29 | <0.001 |
| HDL-C (mg/dl) | 39.73 [13.66] | 39.70 [13.67] | 40.75 [13.47] | -1.1 | -3.7, 1.5 | 0.4 |
| Total cholesterol (mg/dl) | 192.65 [51.91] | 193.29 [51.65] | 174.76 [56.23] | 19 | 7.7, 29 | <0.001 |
| Male (%) | 2,097 (64%) | 2,019 (64%) | 78 (68%) | -4.6% | -14%, 4.5% | 0.4 |

Mean [SD]; n (%)

Welch Two Sample t-test; Two sample test for equality of proportions; 12-sample test for equality of proportions without continuity correction; 14-sample test for equality of proportions without continuity correction

BMI = body mass index; CI = confidence interval; GCS = Glasgow coma scale; HDL-C = high density lipoprotein-cholesterol; LDL-C = low density lipoprotein-cholesterol; LoS = length of stay; NIHSS = National Institute of Health Stroke Scale/Score; n = number of patient; RBS = random blood sugar; SD = standard deviation.

## Predictors for mortality

Table 2(A) and Table 2(B) described the model summary, where both NIHSS score and uric acid level predicted mortality. Male patients had a higher hazard ratio for post-stroke mortality. With a one-year increment, older patients were at a higher hazard of having an event (mortality). This trend was not statistically significant (p = 0.078), with a 95% confidence interval

**Table 2. HR of predictors in causing stroke mortality, after adjusting for age and sex.**

| (a) Fine-Gray Model | | | |
|---|---|---|---|
| **Characteristic** | **HR** | **95% CI** | **p-value** |
| NIHSS (score) | 1.04 | 1.00, 1.07 | 0.037 |
| Cardiovascular Disease (yes/no) | 2.15 | 1.37, 3.37 | <0.001 |
| Renal Disease (yes/no) | 1.38 | 0.87, 2.20 | 0.2 |
| BMI (kg/m$^2$) | 0.94 | 0.89, 0.99 | 0.014 |
| Age (year) | 1.02 | 1.00, 1.04 | 0.077 |
| GCS—Movement | 0.86 | 0.65, 1.15 | 0.3 |
| GCS—Eye | 0.66 | 0.49, 0.89 | 0.007 |
| Sex—Male | 1.52 | 1.02, 2.26 | 0.040 |
| Uric acid (mg/dl) | 1.02 | 1.00, 1.03 | 0.014 |
| Pneumonia (yes/no) | 2.41 | 1.42, 4.09 | 0.001 |
| Sepsis (yes/no) | 2.06 | 1.09, 3.88 | 0.025 |
| (b) Cox's regression model | | | |
| **Characteristic** | **HR** | **95% CI** | **p-value** |
| NIHSS (score) | 1.04 | 1.00, 1.07 | 0.038 |
| Cardiovascular Disease (yes/no) | 2.16 | 1.37, 3.40 | <0.001 |
| Renal Disease (yes/no) | 1.39 | 0.86, 2.23 | 0.2 |
| BMI (kg/m2) | 0.94 | 0.89, 0.99 | 0.015 |
| Age (year) | 1.02 | 1.00, 1.04 | 0.077 |
| GCS—Movement | 0.87 | 0.65, 1.16 | 0.3 |
| GCS—Eye | 0.66 | 0.48, 0.89 | 0.006 |
| Sex—Male | 1.51 | 1.01, 2.26 | 0.044 |
| Uric acid (mg/dl) | 1.02 | 1.00, 1.03 | 0.014 |
| Pneumonia (yes/no) | 2.43 | 1.42, 4.15 | 0.001 |
| Sepsis (yes/no) | 2.07 | 1.09, 3.94 | 0.027 |

BMI = body mass index; CI = confidence interval; GCS = Glasgow coma scale; HR = hazard ratio; NIHSS = National Health Institute Scale/Score

(CI) range of 1.00–1.04. Comorbidities, such as cardiovascular disease, pneumonia, or sepsis, except renal disease, were also strong predictors of in-hospital mortality. In contrast, a higher BMI and GCS would indicate lower hazard of death.

## Discussion

To the best of our knowledge, this is the first large prognostic study on hospital patients with IS from a tertiary care hospital in Indonesia. Our data considerably confirmed that male sex, presence of cardiovascular disease, pneumonia, and sepsis, NIHSS score on admission, hyperuricemia, BMI and GCS were strong predictors of in-hospital mortality in IS patients.

The NBCH is a referral hospital that specializes in stroke and neurological disorders care, which represents the current clinical practice in Indonesia. The mortality rate in our hospital is rather similar to that reported in the United States (4.43% vs. 4.92%). Compared to other countries, the in-hospital mortality rate of our hospital for IS is slightly higher than that of China (4.43% vs. 3.2%) [23], even though the median age of the studied population in both studies was comparable. However, our mortality rate is much lower than that reported for in Sarawak, Malaysia (4.43% vs. 6.7%) [24]. Our study underlined the available comorbidities among IS patients to explore the plausibility of mortality rate differences between our study and reports from other countries.

According to a study conducted in the United States, heart failure and atrial fibrillation are the comorbidities requiring prompt evaluation and revascularization to prevent adverse outcomes among acute IS patients [25]. Patients undergoing intravenous thrombolysis (IVT) had better or comparable in-hospital adverse outcomes than those who did not undergo such a procedure. Prior cardiovascular diseases significantly increase the risk of early mortality in patients with IS and even in those with stroke mimics [26]. Our study confirmed that cardiovascular disease is a significant predictor of in-hospital mortality as well, which could possibly be due to several plausible mechanisms. Patients with cardiovascular disease might also present with heart rate variability (HRV) and autonomic dysfunction, with both predictors increasing the odds of in-hospital stroke mortality. HRV has also been proposed as a biomarker for predicting stroke and its complications or functional outcomes. A previous systematic review suggested that time- and frequency-domain HRV parameters predict post-stroke functionality [27].

Our findings also revealed that hyperuricemia is a significant factor for in-hospital mortality in IS patients. A previous retrospective cohort analysis of 3,731 patients demonstrated that hyperuricemia predicted poor outcomes and is associated with a higher vascular event rate among stroke patients [28]. Hyperuricemia worsens the outcomes of IS by increasing the morbidity and mortality rates. Although the reports on hyperuricemia and stroke outcomes are seemingly inconsistent with conflicting findings, thereby necessitating further investigation. Patho-physiologically, human atherosclerotic plaques contain more urate crystals, which trigger an inflammatory response due to phagocytosis of the deposited urate crystals by polymorphonuclear leukocytes subsequently leading to intimal damage, platelet activation, and coagulation cascades. Moreover, a previous meta-analyses study by Kim et al. showed that the stroke incidence (relative risk (RR) = 1.41) and mortality (RR = 1.36) were strongly associated with high uric acid levels [29].

Obesity, a metabolic disorder, is considered an established predictor of IS [30]. However, many studies suggest a better prognosis among overweight and obese patients after IS [31–33]. The improved survival, functional outcomes and stroke recurrence among overweight and obese IS patients were collectively termed the "obesity stroke paradox" [32, 33]. This appealing evidence was also revealed by our study, wherein a higher BMI was associated with a lower mortality risk. Although the underlying mechanisms are still unclear, the lipid profiles of living patients in our cohort were significantly better than those of patients with events. Furthermore, the ACROSS-China prospective study confirmed that low BMI and low uric acid levels had a combined effect on poor outcomes in IS patients [34].

Although the difference was not statistically significant, men tended to have a higher post-stroke mortality risk. A previous review findings indicated an equal trend of mortality risk between men and women, although several studies in that review revealed a higher prevalence of post-stroke mortality in men [35]. Low post-stroke mortality rates among women have been linked to the protective role of estrogen in preventing stroke. After menopause, the incidence of stroke became similar between men and women. Other studies have also indicated a higher risk of cardiovascular disease and stroke mortality in women with premature or early onset menopause [36]. Willer et al.'s study also revealed that women who were older at the time of stroke had more severe stroke events and more disabilities [20]. Efforts are required to make a more intricate and direct comparison between the sexes.

Studies have also shown a strong correlation between the NIHSS score and higher 30-day mortality risk. One study mentioned the percentages of risk associated with NIHSS score, which were as follows 0–7, 4.2%; 8–13, 13.9%; 14–21, 31.6%; 22–42, 53.5% [37]. In another study reporting the 30-day mortality rate of acute IS patients was 2.3% and >75% for patients with scores of 0 and ≥ 40, respectively. A model with the NIHSS score alone provided nearly

as good discrimination [38] with higher grade indicating a higher risk of mortality. The above data indicated that the NIHSS provides substantial prognostic information regarding the 30-day mortality risk among acute IS cases. This index of stroke severity is a very strong discriminator of mortality risk, even in the absence of other clinical information, regardless of whether it is used as a continuous or categorical risk determinant.

In selecting the predictors of in-hospital mortality in the study, we applied a theoretical-based model, instead of initially using bivariate selection (BVS). Using BVS is inappropriate, despite its common statistical procedure that has been utilized in previous medical research. In BVS, if the statistical $p$ value of a predictor in the bivariate analysis is higher than the arbitrary value (often $p = 0.05$), then this predictor will not be chosen for inclusion in the multivariable analysis [22]. This variable selection technique is inappropriate because the BVS method wrongly rejects potentially plausible variables when the association between a clinical outcome and a predictor is confounded, let alone if this confounder is not appropriately controlled [39]. Data with multiple variables are situated in a higher-dimensional mathematical space and behave differently compared to data in a lower-dimensional space. Consequently, multiple testing in multiple variable data causes an inflation of the error rate [40]; hence, the current statistical literature is strongly against the use of the BVS technique [41]. Although the variable selection method still merits some modelling purposes [42], we opted to follow a theory-driven approach.

Previous publications have highlighted the importance of the theory-driven approach when designing research and analytical procedures [39–42]. Thus, this theoretically driven model explaining in-hospital mortality in IS patients was explored through a multivariable analysis. Furthermore, the analysis was rationalized by experts' opinions on the clinical relevance of the study to formulate a parsimonious model. The final model included the following variables:, male sex, presence of cardiovascular disease, hyperuricemia,, pneumonia, sepsis and NIHSS, BMI, and GCS.

The goal of the present study was to ascertain the survivability of IS patients, and the most widely used statistical methods for survival analysis are the Kaplan–Meier method for estimating survival function and the Cox proportional hazards model for estimating the effects of covariates on the likelihood of the occurrence of in-hospital mortality among patients with IS. The aforementioned methodology typically relies on the assumption of non-informative censoring, which asserts that censoring occurs independently of the risk for the outcome of interest. The violation of this assumption introduces bias in the analysis [43].

Competing risks occur frequently in the survival analysis. A competing risk is an event whose occurrence prohibits the occurrence of clinical outcome of interest [43]. In a study focusing on the time to CVD and stroke mortality, the risk of death attributable to non-CVD and non-stroke causes is called a competing risk. The cumulative incidence function (CIF) and the Fine-Gray model should be used to estimate the crude incidence of clinical outcomes rather than to complement the Kaplan–Meier survival function per se. Given that the estimates of incidence are skewed upward when the Kaplan-Meier survival function is used, whether or not the competing events are related to one another is unknown [44, 45]. To ensure that there is no upward bias in calculating the hazard ratio of in-hospital mortality in IS, we used the CIF and Fine-Gray model in addition to the Kaplan-Meier survival analysis and Cox- regression model in our study (Table 2, Fig 2A and 2B). In fact, the hazard ratios of predictors for in-hospital mortality in IS patients in both models are identical. Hence, our study has no upward bias.

The presence of GCS in predicting mortality was deemed rather straightforward, where patients with a higher GCS score had a better level of consciousness and, thus, better survivability. Among of the three GCS components, the verbal domain was rather difficult to assess

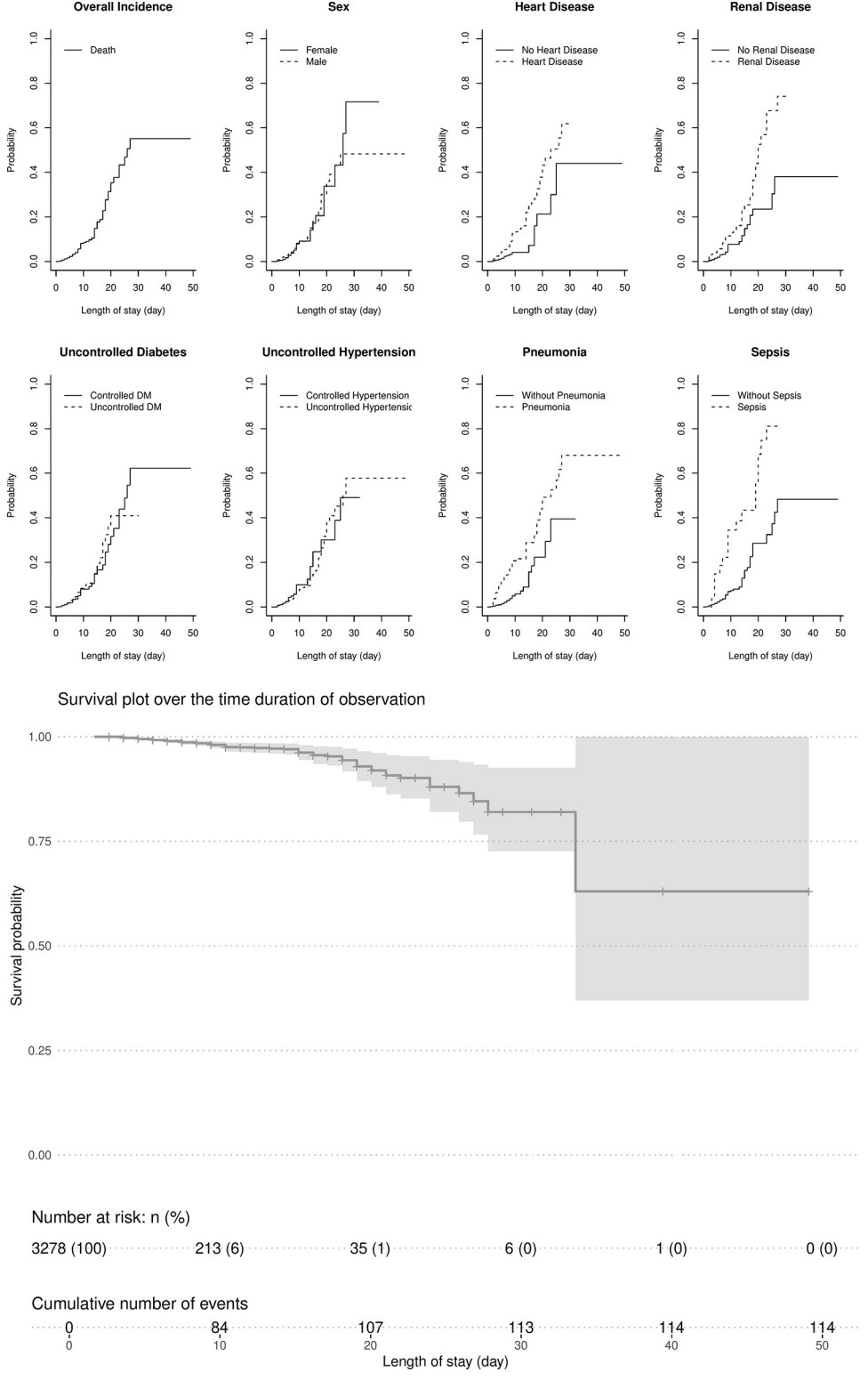

**Fig 2. A.** Cumulative incidence function of stroke mortality, grouped by categorical variable of interest. **B.** Estimated survival probability based on Cox's regression model result in Table 2.

since patients might be verbally incoherent. The GCS score was then separated into the eye, movement, and verbal domains to avoid reduced statistical power due to the presence of mixed statistical errors in the verbal domain. The variability in the verbal domain resulted in its removal during variable selection using stepwise regression.

There are several limitations to the present study. First, our study results could not be generalized to other populations, because they were derived from a single tertiary hospital and only from one year period; the sample size was further limited by omitting patients with TIA, potentially affecting the results of the analysis. Second, our study was conducted in a tertiary referral hospital that specializes in stroke and neurological disorders. Third, the year 2020 was the year COVID-19 pandemic started and it was possible that COVID-19 became one of the risk factors for mortality in our study. However, due to the very small number of COVID-19 patients in our study and none of them died, we could not include COVID-19 diagnosis in the analysis. Therefore, a large population-based study with a nationwide setting and longer period is necessary to confirm our findings.

Regardless of the study limitations, the strength of this study is its ability to capture all the clinical outcomes noted in hospitals among the Indonesian population.

In conclusion, our study data showed that male sex, cardiovascular disease, pneumonia, sepsis, hyperuricemia, NIHSS score on admission and GCS are strong predictors of in-hospital mortality in patients with IS. Additionally, the obesity-stroke paradox, represented by a higher BMI, was confirmed in our cohort.

## Supporting information

**S1 File. Variable selection and statistical analysis.**
(PDF)

## Author Contributions

**Conceptualization:** Nizar Yamanie, Yuli Felistia, Amal Chalik Sjaaf, Muhammad Miftahussurur, Anwar Santoso.

**Data curation:** Nizar Yamanie, Yuli Felistia, Muhammad Miftahussurur, Anwar Santoso.

**Formal analysis:** Yuli Felistia, Nugroho Harry Susanto, Aly Lamuri, Muhammad Miftahussurur, Anwar Santoso.

**Funding acquisition:** Amal Chalik Sjaaf, Muhammad Miftahussurur.

**Investigation:** Nizar Yamanie, Yuli Felistia, Nugroho Harry Susanto, Aly Lamuri, Anwar Santoso.

**Methodology:** Nugroho Harry Susanto, Aly Lamuri, Anwar Santoso.

**Project administration:** Yuli Felistia, Aly Lamuri.

**Resources:** Aly Lamuri, Amal Chalik Sjaaf, Muhammad Miftahussurur, Anwar Santoso.

**Software:** Nugroho Harry Susanto, Aly Lamuri.

**Supervision:** Nugroho Harry Susanto, Aly Lamuri, Amal Chalik Sjaaf, Muhammad Miftahussurur, Anwar Santoso.

**Validation:** Nugroho Harry Susanto, Muhammad Miftahussurur.

**Visualization:** Nugroho Harry Susanto, Aly Lamuri, Amal Chalik Sjaaf, Anwar Santoso.

**Writing – original draft:** Nizar Yamanie, Yuli Felistia, Aly Lamuri.

**Writing – review & editing:** Nizar Yamanie, Nugroho Harry Susanto, Muhammad Miftahus-surur, Anwar Santoso.

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
