## [Decision Letter · Decision Letter 0]

5 Jul 2023

PONE-D-23-03260Prognostic Model of In-hospital Ischemic Stroke Mortality Based on an Electronic Health Record Cohort in  IndonesiaPLOS ONE

Dear Dr. Santoso,

Thank you for submitting your manuscript to PLOS ONE. After careful consideration, we feel that it has merit but does not fully meet PLOS ONE’s publication criteria as it currently stands. Therefore, we invite you to submit a revised version of the manuscript that addresses the points raised during the review process. Please submit your revised manuscript by Aug 19 2023 11:59PM. If you will need more time than this to complete your revisions, please reply to this message or contact the journal office at plosone@plos.org. Please include the following items when submitting your revised manuscript:A rebuttal letter that responds to each point raised by the academic editor and reviewer(s). You should upload this letter as a separate file labeled 'Response to Reviewers'.A marked-up copy of your manuscript that highlights changes made to the original version. You should upload this as a separate file labeled 'Revised Manuscript with Track Changes'.An unmarked version of your revised paper without tracked changes. You should upload this as a separate file labeled 'Manuscript'.

We look forward to receiving your revised manuscript.

Kind regards,

Yee Gary Ang, MBBS MPH

Academic Editor

PLOS ONE

Journal Requirements:

1) Tian, X., Liu, J., Yu, C., Hou, Y., Zhan, C., Lin, Q., ... & Wang, J. (2021). Long-term trends in stroke management and burden among low-income women in a rural area from China (1992–2019): a prospective population-based study. Frontiers in Neurology, 1895.

2) Fonarow GC, Saver JL, Smith EE, Broderick JP, Kleindorfer DO, Sacco RL, Pan W, Olson DM, Hernandez AF, Peterson ED, Schwamm LH. Relationship of national institutes of health stroke scale to 30-day mortality in medicare beneficiaries with acute ischemic stroke. J Am Heart Assoc. 2012 Feb;1(1):42-50. doi: 10.1161/JAHA.111.000034.

3) Sun GW, Shook TL, Kay GL. Inappropriate use of bivariable analysis to screen risk factors for use in multivariable analysis. J Clin Epidemiol. 1996 Aug;49(8):907-16. doi: 10.1016/0895-4356(96)00025-x. PMID: 8699212.

In your revision ensure you cite all your sources (including your own works), and quote or rephrase any duplicated text outside the methods section. Further consideration is dependent on these concerns being addressed.

5. Please ensure that you refer to Figures 1 and 2 in your text as, if accepted, production will need this reference to link the reader to the figure.

**Additional Editor Comments:**

This is an interesting study and I invite the authors to submit a revised version addressing the concerns of the 2 reviewers. 

Reviewers' comments:

Reviewer's Responses to Questions

**Comments to the Author**

1. Is the manuscript technically sound, and do the data support the conclusions?

Reviewer #1: Yes

Reviewer #2: Partly

2. Has the statistical analysis been performed appropriately and rigorously? 

Reviewer #1: Yes

Reviewer #2: No

3. Have the authors made all data underlying the findings in their manuscript fully available?

Reviewer #1: Yes

Reviewer #2: Yes

4. Is the manuscript presented in an intelligible fashion and written in standard English?

Reviewer #1: Yes

Reviewer #2: Yes

5. Review Comments to the Author

Reviewer #1: thank you for the review chance. I would like to recommend the inclusion of thrombophilia types [e.g. protein C, protein S, antithrombin deficiency ..etc] as well as inclusion of other prognostic scales " e.g. THRIVE and Orpington scales...etc" as the Asian population has higher incidence of small-vessel ischemic stroke and stroke occurrence in younger ages as compared to Caucasian population, partly due to thrombophilia etiologies.

Reviewer #2: In this manuscript entitled 'Prognostic Model of In-hospital Ischemic Stroke Mortality Based on an Electronic Health Record Cohort in Indonesia' the authors investigate the predictors of in-hospital mortality in patients with ischemic stroke (IS). The study is based on the IS medical records from the National Brain Centre Hospital, Jakarta, Indonesia with a total of 3278 patients with IS. The authors worked on an interesting topic regarding public health meassage and local regional South-east Asia countries but, in my opinion, there are some methodological issues and suggestions to consider.

- there is a discrepancy between the period in which the study was conducted (one would imagine operationally) and the period of observation and date collection, respectively October 2021 to February 2022 versus years 2019-2020 (without specifying exact beginning and end). The authors should better explain this difference and possibly consider the possibility, if any, of extending the observation period. It would also be useful for the date collection to have a precise observation period (not only year but also month and/or day).

- As far as the outcome under analysis is concerned, it is presumable that the cases were identified using the ICD definition. This should be made explicit.

- It is unclear whether the mortality assessment was only carried out intra-hospital. Did the observation continue for some time outside the hospital as well?

- The time-to-event analysis should be enriched with a competitive risk analysis as discharge represents an event that can drastically change the patients' medical history. Therefore not only simple COX but also cumulative incidence function curves and the Fine and Gray model (Resche-Rigon M et al. Crit Care. 2006 Feb;10(1):R5).

- The authors claim to have removed all records with missing data. Before simple removal it would be appropriate to investigate the significance of this missingness using appropriate analyses (pattern analysis, missing per patient, missing per item, missing by outcome). It is evident that, for example, a reduction in the number of subjects leads to a change in the incidence of outcomes (from 4.69% mortality to 3.48%), so it is assumed that in the 7.9% of missing data there is a different percentage of subjects with an unfavourable outcome. Could this have introduced a bias into the analysis?

- The authors state that they selected the variables using a stepwise regression, forward or backward? Furthermore, they mention the use of other classification models compared with appropriate metrics. Both models and metrics should be listed and in a hypothetical supplementary material, the results of these comparisons should be presented to see if the type of algorithm may have influenced the final selection of variables.

- Table 1 should also present the overall data and not only the comparison between subjects with and without an event.

- Was the GCS used as a continuous variable or category? Shouldn't it be a continuous score?

- The figure presented represents the survival plot as estimated from the Cox's regression model. The time-to-event variable should be described a priori without any modelling. The overall kaplan meier, median follow-up, median survival and confidence interval should be presented. This would help readers to better understand the type of patients and the time frame studied.

6. PLOS authors have the option to publish the peer review history of their article (what does this mean?). If published, this will include your full peer review and any attached files.

Reviewer #1: **Yes: **Mohamed Mostafa

Reviewer #2: No

---

## [Author Response · Author response to Decision Letter 0]

18 Aug 2023

Response to Reviewers

Reviewer #1: thank you for the review chance. I would like to recommend the inclusion of thrombophilia types [e.g. protein C, protein S, antithrombin deficiency ..etc] as well as inclusion of other prognostic scales " e.g. THRIVE and Orpington scales...etc" as the Asian population has higher incidence of small-vessel ischemic stroke and stroke occurrence in younger ages as compared to Caucasian population, partly due to thrombophilia etiologies.

We thank the reviewer for this important comment and agree that the suggested prognostic scales and thrombophilia types are important for Asian population. Sadly, we do not have the necessary information. Hence, we did not include nor mention any of those data in our manuscript.

Reviewer #2: In this manuscript entitled 'Prognostic Model of In-hospital Ischemic Stroke Mortality Based on an Electronic Health Record Cohort in Indonesia' the authors investigate the predictors of in-hospital mortality in patients with ischemic stroke (IS). The study is based on the IS medical records from the National Brain Centre Hospital, Jakarta, Indonesia with a total of 3278 patients with IS. The authors worked on an interesting topic regarding public health meassage and local regional South-east Asia countries but, in my opinion, there are some methodological issues and suggestions to consider. 

- there is a discrepancy between the period in which the study was conducted (one would imagine operationally) and the period of observation and date collection, respectively October 2021 to February 2022 versus years 2019-2020 (without specifying exact beginning and end). The authors should better explain this difference and possibly consider the possibility, if any, of extending the observation period. It would also be useful for the date collection to have a precise observation period (not only year but also month and/or day).

We acknowledge that the statement of operational study duration and data collection duration led to confusion. To avoid this problem, we removed the line about operational study duration in paragraph 1 of methods section. We apologize for the wrong data collection period written in paragraph 4 of methods section, we changed the data collection years in the line and added the specific start and end date of data collection duration.

Old line: “Data for a period of one year were collected from 2019 to 2020.”

New line: “Data for a period of one year was collected from 1 January to 31 December 2020.”

 - As far as the outcome under analysis is concerned, it is presumable that the cases were identified using the ICD definition. This should be made explicit.

We realize that we did not write it clearly that the diagnosed cases were recorded in medical record along with their ICD definition. We revised the following line in the first paragraph of methods section:

Old line: “The study population included hospitalised IS patients according to the Trial of Org 10172 in Acute Stroke Treatment (TOAST) definition.”

New line: “The study population included hospitalised IS patients according to the Trial of Org 10172 in Acute Stroke Treatment (TOAST) definition along with the ICD-10 code I63 for acute ischemic stroke.”

- It is unclear whether the mortality assessment was only carried out intra-hospital. Did the observation continue for some time outside the hospital as well?

Our apologies for not stated it in the methods section that we only assessed mortality during hospitalization. We added the necessary information in the first line of Clinical outcomes of methods section.

Old line: “The clinical outcomes were all-cause and stroke-related mortality rates.”

New line: “The clinical outcomes were all-cause and stroke-related mortality rates during hospitalization.”

 - The time-to-event analysis should be enriched with a competitive risk analysis as discharge represents an event that can drastically change the patients' medical history. Therefore not only simple COX but also cumulative incidence function curves and the Fine and Gray model (Resche-Rigon M et al. Crit Care. 2006 Feb;10(1):R5).

We thank the reviewer for raising the issue about competing risk. We added the cumulative incidence function curves and the comparison of Cox regression results against Fine and Gray models in the results section.

 - The authors claim to have removed all records with missing data. Before simple removal it would be appropriate to investigate the significance of this missingness using appropriate analyses (pattern analysis, missing per patient, missing per item, missing by outcome). It is evident that, for example, a reduction in the number of subjects leads to a change in the incidence of outcomes (from 4.69% mortality to 3.48%), so it is assumed that in the 7.9% of missing data there is a different percentage of subjects with an unfavourable outcome. Could this have introduced a bias into the analysis?

Indeed, we took into account the possibility of bias being introduced in this analysis. We thoroughly explored the missingness to assess the mechanism, i.e. MCAR/MAR/MNAR. It’s understandably challenging to differentiate MAR/MNAR through statistical analysis procedure; and it is more practical to determine whether the missing values occurred due to MCAR/non-MCAR, which could be MAR/MNAR. For clinical research, complete-case analysis in datasets with MAR entries still produces valid results (read more: https://doi.org/10.1016/j.jclinepi.2022.08.016). In another review of published RCTs, the majority of published studies still used complete case-analysis by discarding entries with null values (read more: https://doi.org/10.1186/1471-2288-14-118). Meanwhile, another review also suggested excluding observation in case of missingness in primary predictors for studies with cross-sectional design (read more: https://doi.org/10.1016/j.pmrj.2015.07.011). As such, we concluded that using a complete-case analysis is still arguably sound and justified. We attached our exploration on missingness in our supplementary materials, page 2-12.

- The authors state that they selected the variables using a stepwise regression, forward or backward? Furthermore, they mention the use of other classification models compared with appropriate metrics. Both models and metrics should be listed and in a hypothetical supplementary material, the results of these comparisons should be presented to see if the type of algorithm may have influenced the final selection of variables.

We agree that more detail about the analysis should be added as supplementary. We attached the supplementary about the detail of our analysis along with our revised manuscript.

We used a bidirectional stepwise regression, a combination of forward and backward selection. We added this information in the first paragraph of Statistical analysis. 

Old line: “To aid the exploratory procedure, significant variables reported in previous studies are included in stepwise regression models.”

New line: “To aid the exploratory procedure, significant variables reported in previous studies are included in stepwise regression models using a combination of forward and backward selection.”

- Table 1 should also present the overall data and not only the comparison between subjects with and without an event.

We added a column in Table 1 that showed the baseline characteristic for the overall patients in our manuscript. 

- Was the GCS used as a continuous variable or category? Shouldn't it be a continuous score?

We understand that the GCS should be a continuous variable however the clinicians only assessed eye and movement scores of GCS as continuous measurement while the verbal as either coherence or incoherence. Thus, we decided to include GCS in the analysis separately for eye, movement, and verbal assessment.

- The figure presented represents the survival plot as estimated from the Cox's regression model. The time-to-event variable should be described a priori without any modelling. The overall kaplan meier, median follow-up, median survival and confidence interval should be presented. This would help readers to better understand the type of patients and the time frame studied.

Thank you for the suggestion, we have added the descriptive statistics in our supplementary file from page 22 to 29.

---

## [Decision Letter · Decision Letter 1]

16 Oct 2023

PONE-D-23-03260R1Prognostic Model of In-hospital Ischemic Stroke Mortality Based on an Electronic Health Record Cohort in  IndonesiaPLOS ONE

Dear Dr. Santoso,

Thank you for submitting your manuscript to PLOS ONE. After careful consideration, we feel that it has merit but does not fully meet PLOS ONE’s publication criteria as it currently stands. Therefore, we invite you to submit a revised version of the manuscript that addresses the points raised during the review process.

We look forward to receiving your revised manuscript.

Kind regards,

Yee Gary Ang, MBBS MPH

Academic Editor

PLOS ONE

Additional Editor Comments:

The original reviewers were not available to review the paper so we had asked 2 more reviewers. Please kindly address their concerns before resubmitting.

Reviewers' comments:

Reviewer's Responses to Questions

**Comments to the Author**

1. If the authors have adequately addressed your comments raised in a previous round of review and you feel that this manuscript is now acceptable for publication, you may indicate that here to bypass the “Comments to the Author” section, enter your conflict of interest statement in the “Confidential to Editor” section, and submit your "Accept" recommendation.

Reviewer #3: (No Response)

Reviewer #4: (No Response)

2. Is the manuscript technically sound, and do the data support the conclusions?

Reviewer #3: Partly

Reviewer #4: Yes

3. Has the statistical analysis been performed appropriately and rigorously? 

Reviewer #3: I Don't Know

Reviewer #4: Yes

4. Have the authors made all data underlying the findings in their manuscript fully available?

Reviewer #3: Yes

Reviewer #4: Yes

5. Is the manuscript presented in an intelligible fashion and written in standard English?

Reviewer #3: No

Reviewer #4: Yes

6. Review Comments to the Author

Reviewer #3: The authors present a retrospective observational study used IS medical records from a single (teriary, high volume) center in Indonesia, with predictors of in-hospital mortality in patients with IS.

I would suggest the authors to consider the following suggestions:

- having the manuscript checked by a native english speaker for errors regarding grammar, synthax etc.

- having a more focused and structured discussion section

Reviewer #4: The author explore in their paper Prognostic Model of In-hospital Ischemic Stroke Mortality Based on an Electronic Health Record Cohort in Indonesia.

The research is intriguing however i have some concern about the possibile confounding variabels

Major Criticism

Do the authors explore in deep all cause of mortality? Indeed Ischemic Stroke has several complications, including infections?

indeed mortality after ischemic stroke other than metabolic or cardiac causes has other factors as infection well known since a decades: "Post-stroke infection: A systematic review and meta-analysis 10.1186/1471-2377-11-110" particularly pneumonia "

Int J Stroke. 2019 Feb;14(2):125-136. doi: 10.1177/1747493018806196"

Therefore please specify if you can have access or not to ICD classification on Clicical dismission records

Further it would be usefull also estimate how many patients had previous hosptial access

Minor Criticism

improve material and methods better clarifing the data extraction emthofd and model on ICD evaluation, for istance you shoudl eport waht kind of code you explored and why you selected those

Improve tables in terms of better details about previous hospital access and cause of mortality

7. PLOS authors have the option to publish the peer review history of their article (what does this mean?). If published, this will include your full peer review and any attached files.

Reviewer #3: No

Reviewer #4: No

---

## [Author Response · Author response to Decision Letter 1]

14 Dec 2023

Response to Reviewers 

Reviewer #3: The authors present a retrospective observational study used IS medical records from a single (tertiary, high volume) center in Indonesia, with predictors of in-hospital mortality in patients with IS.

I would suggest the authors to consider the following suggestions:

- having the manuscript checked by a native English speaker for errors regarding grammar, synthax etc.

- having a more focused and structured discussion section.

We thanked reviewer for the comment. We revised discussion section to make the structure better and more focused. Further, we asked professional proofreader to proofread our manuscript after we addressed comments from all reviewers. 

Reviewer #4: Major Criticism

Do the authors explore in deep all cause of mortality? Indeed, Ischemic Stroke has several complications, including infections?

indeed mortality after ischemic stroke other than metabolic or cardiac causes has other factors as infection well known since a decade: "Post-stroke infection: A systematic review and meta-analysis 10.1186/1471-2377-11-110" particularly pneumonia "

Int J Stroke. 2019 Feb;14(2):125-136. doi: 10.1177/1747493018806196"

Therefore please specify if you can have access or not to ICD classification on Clinical dismission records

Further it would be useful also estimate how many patients had previous hospital access.

Minor Criticism

improve material and methods better clarifying the data extraction method and model on ICD evaluation, for instance you should report what kind of code you explored and why you selected those.

Improve tables in terms of better details about previous hospital access and cause of mortality.

We appreciated the reviewer’s inputs to improve our manuscripts. We went back to our medical record to get the related data, however due to time constraint we could only get aggregate data. It was also very hard for us to get specific ICD 10 for cause of death because clinical autopsy is not compulsory procedure in Indonesia.

499 (14%) patients were referred from other hospital and the prominent cause of death was cardiac arrest, followed by pneumonia, and sepsis. We decided to add these data in the first paragraph of results section, not in table 1 because we only have aggregate data.

The following lines were added the first paragraph of results section:

Around 14% of the patients (499 out of 3,561) were referred from other previous hospitals. The top three causes of death were cardiac arrest, pneumonia, and sepsis.

---

## [Decision Letter · Decision Letter 2]

20 Feb 2024

PONE-D-23-03260R2Prognostic Model of In-hospital Ischemic Stroke Mortality Based on an Electronic Health Record Cohort in  IndonesiaPLOS ONE

Dear Dr. Santoso,

Thank you for submitting your manuscript to PLOS ONE. After careful consideration, we feel that it has merit but does not fully meet PLOS ONE’s publication criteria as it currently stands. Therefore, we invite you to submit a revised version of the manuscript that addresses the points raised during the review process.

There were mixed reviews between 4 reviewers. Please kindly read through the comments carefullly and address them before submitting again. 

We look forward to receiving your revised manuscript.

Kind regards,

Yee Gary Ang, MBBS MPH

Academic Editor

PLOS ONE

Reviewers' comments:

Reviewer's Responses to Questions

**Comments to the Author**

1. If the authors have adequately addressed your comments raised in a previous round of review and you feel that this manuscript is now acceptable for publication, you may indicate that here to bypass the “Comments to the Author” section, enter your conflict of interest statement in the “Confidential to Editor” section, and submit your "Accept" recommendation.

Reviewer #3: (No Response)

Reviewer #5: All comments have been addressed

Reviewer #6: All comments have been addressed

Reviewer #7: (No Response)

2. Is the manuscript technically sound, and do the data support the conclusions?

Reviewer #3: Partly

Reviewer #5: Yes

Reviewer #6: Yes

Reviewer #7: Partly

3. Has the statistical analysis been performed appropriately and rigorously? 

Reviewer #3: I Don't Know

Reviewer #5: Yes

Reviewer #6: Yes

Reviewer #7: No

4. Have the authors made all data underlying the findings in their manuscript fully available?

Reviewer #3: Yes

Reviewer #5: Yes

Reviewer #6: Yes

Reviewer #7: No

5. Is the manuscript presented in an intelligible fashion and written in standard English?

Reviewer #3: No

Reviewer #5: Yes

Reviewer #6: No

Reviewer #7: Yes

6. Review Comments to the Author

Reviewer #3: The most important limitation of this study, besides the lack of generalizability, is the completeness of the data. This results in omitting important prognostic factors such as glycemia, history of diabetes and occurence of infection. Although pneumonia and sepsis were in the top 3 causes of mortality, presence of infection was not included in the model.

Reviewer #5: (No Response)

Reviewer #6: The authors raise a very important socio-economic public health problem stroke. However, the prediction model nowadays are computed using machine learning methods this can not render the merit for the publication of this study given lack of this kind of studies in LMICs.

The English language must be improved!!

Reviewer #7: Thank you for your hard work conducting the research.

The study findings provide valuable insights into predictors of in-hospital mortality among IS patients in Indonesia.

Here are a few comments:

1. Subject Selection Process

The process of selecting subjects should be presented more clearly than the variable extraction process to reduce selection bias. Exclusion criteria should be defined and the reasons for exclusion should be explained. It should also be clearly indicated how many subjects were excluded in the process. Based on the information provided in the text alone, it is difficult to determine which group was ultimately included in the analysis. Especially, since there were quite a few subjects without blood glucose data, it is unclear whether they were all excluded from the analysis.

If the target group is not transparently defined, the results are difficult to trust as they may reflect analysis of a heterogeneous population.

2. Adjustment for Comorbidities

Adjustment for comorbidities is necessary using a comorbidity index. If obtaining variables for the comorbidity index is difficult, it is necessary to investigate and mention the major comorbidities. Adjustment should be made based on the number of comorbidities, or alternatively, adjustment should be made for major comorbidities.

Especially, since the data was collected during the Covid-19 pandemic, Covid-19-related deaths could have had a significant impact, so this should be taken into consideration.

3. Co-variants Definition

There is no accurate definition for heart disease, renal disease, uncontrolled diabetes, uncontrolled hypertension, etc.

4. Sex as Predictors for Mortality

If sex becomes an important factor, additional analysis should be conducted by dividing by gender to determine if there are differences in factors related to post-stroke mortality according to sex.

5. Figure 2 Needs Modification

It seems that Figure 2 needs to be revised. Is the x-axis in Figure 2(A) correct? Is it "year" or "length of stay"? Although it is not an observation for 50 years, the graph indicates that it has been tracked for 50 years.

The intended message in Figure 2(B) is not well understood. It's not clear what is being explained in the survival analysis. It would be helpful if the rationale for showing the survival curve in the manuscript is well explained.

6. Other Comments

1) The lack of explanations for tables and figures in the text makes it difficult to match them with the content and lowers comprehensibility.

2) It is necessary to mention the possibility of more intensive drug therapy for patients with severe conditions who had low blood triglyceride levels, low-density lipoprotein (LDL) levels, and total cholesterol.

3) It should be noted that the analyzed data represent only one year and may not reflect long-term trends, and that the data collected from the hospital may be limited in terms of external environmental factors or additional factors affecting patients.

7. PLOS authors have the option to publish the peer review history of their article (what does this mean?). If published, this will include your full peer review and any attached files.

Reviewer #3: No

Reviewer #5: No

Reviewer #6: **Yes: **Amanuel Sisay Endeshaw

Reviewer #7: No

---

## [Author Response · Author response to Decision Letter 2]

28 Apr 2024

Response to reviewers:

1. Subject Selection Process

The process of selecting subjects should be presented more clearly than the variable extraction process to reduce selection bias. Exclusion criteria should be defined and the reasons for exclusion should be explained. It should also be clearly indicated how many subjects were excluded in the process. Based on the information provided in the text alone, it is difficult to determine which group was ultimately included in the analysis. Especially, since there were quite a few subjects without blood glucose data, it is unclear whether they were all excluded from the analysis.

If the target group is not transparently defined, the results are difficult to trust as they may reflect analysis of a heterogeneous population.

We thank the reviewer for highlighting the problem in understanding the patient’s selection process for our analysis. We replaced figure 1 with STROBE flowchart to make it easier to understand how we came to the final number of 3,278 patients for the analysis. 

2. Adjustment for Comorbidities

Adjustment for comorbidities is necessary using a comorbidity index. If obtaining variables for the comorbidity index is difficult, it is necessary to investigate and mention the major comorbidities. Adjustment should be made based on the number of comorbidities, or alternatively, adjustment should be made for major comorbidities.

Especially, since the data was collected during the Covid-19 pandemic, Covid-19-related deaths could have had a significant impact, so this should be taken into consideration.

We agree that comorbidities are important risk factors. We added pneumonia and sepsis into the analysis and revised the tables, results, and discussion to accommodate the new results.

However, we could not add COVID-19 diagnosis as one of comorbidities into the analysis because only 13 patients were diagnosed with COVID-19 and none of them died. We added this into the limitation of our study.

3. Co-variants Definition

There is no accurate definition for heart disease, renal disease, uncontrolled diabetes, uncontrolled hypertension, etc.

We apologize that we did not write a clear definition for uncontrolled diabetes and uncontrolled hypertension. We added the definition for those 2 terms under the paragraphs (highlighted in yellow) for the definition of heart disease and renal disease. Additionally, diagnosis of cardiovascular disease and renal disease were determined using the International Classification of Diseases, 10th revision (ICD-10) as a standard operational procedure in our hospital, as presented in Methods section.

4. Sex as Predictors for Mortality

If sex becomes an important factor, additional analysis should be conducted by dividing by gender to determine if there are differences in factors related to post-stroke mortality according to sex.

Thanks indeed for your suggestion, and in addressing your valuable suggestion we already performed the Cox-regression model and Fine-gray model as well, to adjust the confounders. These were presented in Table-2 A and table-2B. 

5. Figure 2 Needs Modification

It seems that Figure 2 needs to be revised. Is the x-axis in Figure 2(A) correct? Is it "year" or "length of stay"? Although it is not an observation for 50 years, the graph indicates that it has been tracked for 50 years.

The intended message in Figure 2(B) is not well understood. It's not clear what is being explained in the survival analysis. It would be helpful if the rationale for showing the survival curve in the manuscript is well explained.

Once again, we thank the reviewer for pointing out the mistake in Figure 2A. The correct x-axis should be “length of stay”, not “year”. We replaced the figure with the correct one.

We agree with the reviewer that the title for Figure 2B was not clear enough.

We changed the title into:

Figure 2B Estimated survival probability based on Cox’s regression model result in table 2B.

6. Other Comments

1) The lack of explanations for tables and figures in the text makes it difficult to match them with the content and lowers comprehensibility.

2) It is necessary to mention the possibility of more intensive drug therapy for patients with severe conditions who had low blood triglyceride levels, low-density lipoprotein (LDL) levels, and total cholesterol.

3) It should be noted that the analyzed data represent only one year and may not reflect long-term trends, and that the data collected from the hospital may be limited in terms of external environmental factors or additional factors affecting patients.

Thanks indeed for your valuable suggestions 

• We already added the explanation within the text (Results section#) to improve comprehensibility as suggested. 

• The treatment strategy was applied in out hospital based on the clinical practice guidelines, particularly for treating dyslipidemia. 

• We added into limitation that our data could not be generalized to other populations for multiple reasons, including that it only represented 1 year period.

---

## [Decision Letter · Decision Letter 3]

16 May 2024

PONE-D-23-03260R3Prognostic Model of In-hospital Ischemic Stroke Mortality Based on an Electronic Health Record Cohort in  IndonesiaPLOS ONE

Dear Dr. Santoso,

Thank you for submitting your manuscript to PLOS ONE. After careful consideration, we feel that it has merit but does not fully meet PLOS ONE’s publication criteria as it currently stands. Therefore, we invite you to submit a revised version of the manuscript that addresses the points raised during the review process.

< Please address the residual comments before resubmitting. />==============================

We look forward to receiving your revised manuscript.

Kind regards,

Yee Gary Ang, MBBS MPH

Academic Editor

PLOS ONE

Journal Requirements:

Reviewers' comments:

Reviewer's Responses to Questions

**Comments to the Author**

1. If the authors have adequately addressed your comments raised in a previous round of review and you feel that this manuscript is now acceptable for publication, you may indicate that here to bypass the “Comments to the Author” section, enter your conflict of interest statement in the “Confidential to Editor” section, and submit your "Accept" recommendation.

Reviewer #7: All comments have been addressed

Reviewer #8: All comments have been addressed

2. Is the manuscript technically sound, and do the data support the conclusions?

Reviewer #7: Partly

Reviewer #8: Partly

3. Has the statistical analysis been performed appropriately and rigorously? 

Reviewer #7: I Don't Know

Reviewer #8: Yes

4. Have the authors made all data underlying the findings in their manuscript fully available?

Reviewer #7: No

Reviewer #8: Yes

5. Is the manuscript presented in an intelligible fashion and written in standard English?

Reviewer #7: Yes

Reviewer #8: Yes

6. Review Comments to the Author

Reviewer #7: Thank you for your hard work.

It's not easy to determine which group each graph represents in Figure 2-A. It would be helpful to specify the names of the groups for the dotted and solid line graphs.

Please label the X-axis in Figure 2-B. It is likely to be "length of day."

Reviewer #8: The answers given to the questions asked in the previous evaluations and the requested corrections were adequately fulfilled.

7. PLOS authors have the option to publish the peer review history of their article (what does this mean?). If published, this will include your full peer review and any attached files.

Reviewer #7: No

Reviewer #8: No

---

## [Author Response · Author response to Decision Letter 3]

23 May 2024

Dear Reviewers,

Thank you for your valuable feedback on our manuscript. We have carefully considered your comments and made the necessary revisions to improve the clarity and accuracy of our paper.

Specifically, you mentioned the need for labeling the X-axis in Figure 2-B. We have addressed this concern and have now labeled the X-axis as "Length of Day." We also have revised label of predictors in Figure-2A. This label accurately reflects the data presented in the figure and ensures that the information is clear and interpretable for our readers. 

Additionally, we have incorporated some useful information and enhanced the clarity of the manuscript based on your suggestions. 

We appreciate your attention to detail and believe that these revisions enhance the overall quality of our manuscript. Please find the revised figure included in the updated manuscript.

Thank you once again for your insightful suggestions.

Sincerely,

---

## [Editor Report · Decision Letter 4]

24 May 2024

Prognostic Model of In-hospital Ischemic Stroke Mortality Based on an Electronic Health Record Cohort in  Indonesia

PONE-D-23-03260R4

Dear Dr. Santoso,

We’re pleased to inform you that your manuscript has been judged scientifically suitable for publication and will be formally accepted for publication once it meets all outstanding technical requirements.

Kind regards,

Yee Gary Ang, MBBS MPH

Academic Editor

PLOS ONE
---

## [Editor Report · Acceptance letter]

3 Jun 2024

PONE-D-23-03260R4 

PLOS ONE

Dear Dr. Santoso, 

I'm pleased to inform you that your manuscript has been deemed suitable for publication in PLOS ONE. Congratulations! Your manuscript is now being handed over to our production team.

Kind regards, 

on behalf of

Dr. Yee Gary Ang 

Academic Editor

PLOS ONE